# Characteristics of the Genome, Transcriptome and Ganoderic Acid of the Medicinal Fungus *Ganoderma lingzhi*

**DOI:** 10.3390/jof8121257

**Published:** 2022-11-28

**Authors:** Qiang Wu, Huan Liu, Yixin Shi, Wanting Li, Jia Huang, Feifei Xue, Yongnan Liu, Gaoqiang Liu

**Affiliations:** 1Hunan Provincial Key Laboratory of Forestry Biotechnology, Central South University of Forestry & Technology, Changsha 410004, China; 2International Cooperation Base of Science and Technology Innovation on Forest Resource Biotechnology of Hunan Province, Central South University of Forestry & Technology, Changsha 410004, China; 3Microbial Variety Creation Center, Yuelushan Laboratory of Seed Industry, Changsha 410004, China

**Keywords:** *Ganoderma lingzhi*, whole genome characteristics, ganoderic acid, RNA sequencing, alternative splicing

## Abstract

*Ganoderma* (Ganodermaceae) is a genus of edible and medicinal mushrooms that create a diverse set of bioactive compounds. *Ganoderma lingzhi* has been famous in China for more than 2000 years for its medicinal properties. However, the genome information of *G. lingzhi* has not been characterized. Here, we characterized its 49.15-Mb genome, encoding 13,125 predicted genes which were sequenced by the Illumina and PacBio platform. A wide spectrum of carbohydrate-active enzymes, with a total number of 519 CAZymes were identified in *G. lingzhi*. Then, the genes involved in sexual recognition and ganoderic acid (GA, key bioactive metabolite) biosynthesis were characterized. In addition, we identified and deduced the possible structures of 20 main GA constituents by UPLC-ESI-MS/MS, including a new special ganochlearic acid A. Furthermore, 3996 novel transcripts were discovered, and 9276 genes were predicted to have the possibility of alternative splicing from RNA-Seq data. The alternative splicing genes were enriched for functional categories involved in protein processing, endocytosis, and metabolic activities by KEGG. These genomic, transcriptomic, and GA constituents’ resources would enrich the toolbox for biological, genetic, and secondary metabolic pathways studies in *G. lingzhi*.

## 1. Introduction

*Ganoderma* (Ganodermaceae) is a genus of edible and medicinal mushrooms that are widely distributed in the tropic and subtropics of Asia, Africa, and America. The medicinal values of this mushroom were documented thousands of years ago [1]. The *G. lingzhi* species is a mushroom that has been renowned in China, which has been incorrectly considered to be another species, *G. lucidum*, which is distributed in Europe, for many years. The most striking characteristics which differentiate *G. lingzhi* from *G. lucidum* are the presence of melanoid bands in the context, a yellow pore surface and thick dissepiments at maturity by morphological studies, rDNA nuc-ITS sequences, and additional gene fragments analysis [2]. The attractive characteristic of *Ganoderma* is its immunomodulatory and anti-tumor activities, which are mainly attributed to the major active components, ganoderic acids (GAs), a type of triterpenoids [3]. GAs are effective as supplemental therapies and improve health when combined with other medications to treat hepatitis, fatigue syndrome, and prostate cancer [4]. Up to now, there are four whole genome sequencing projects for *G. lucidum* (Accession no. PRJDA61381, *G. lucidum* strain BCRC 37177, whole genome shotgun sequences available; PRJNA77007 *G. lucidum* strain Xiangnong No. 1, high-quality genome assembly available; PRJNA71455 *G. lucidum* strain G.260125-1, high-quality genome assembly available; *G. lucidum* strain Ling-Jian No. 2, high-quality genome assembly available) have been published and registered in the National Center of Biotechnology Information (NCBI) [5,6,7,8]. Recently, with the combination of Illumina and PacBio sequencing strategies, we de novo sequenced and assembled the genome of *G. lingzhi* [9]. However, the whole genome information of *G. lingzhi* has not been well characterized yet.

In this work, the genes involved in lignocellulose degradation and sexual recognition, and ganoderic acid biosynthesis of the *G. lingzhi* genome were characterized and analyzed. Then, the key bioactive metabolite, ganoderic acid, was identified by mass spectrometry. Furthermore, novel transcripts were discovered and differential alternative splicings were detected from RNA-Seq data. Collectively, these foundational genomic, transcriptomic, and ganoderic acid bioactive compound resources would enrich the toolbox for further biological, genetic, and secondary metabolic pharmacological studies in *G. lingzhi*.

## 2. Materials and Methods

### 2.1. Fungal Strains and Culture Conditions

The *Ganoderma lingzhi* strain SCIM 1006 (No. CGMCC 18819) kindly provided by Prof. Yu-Cheng Dai was selected for characteristics of the whole genome information, identification of ganoderic acids (GAs), and RNA-Seq [2]. The *G. lingzhi* strain was cultured at 27 °C with shaking at 160 rpm in the dark on an artificial medium (glucose, 44.0; corn flour, 0.5; peptone, 6.5; KH_2_PO_4_, 0.75; MgSO_4_·7H_2_O, 0.45; vitamin B1, 0.01; in g/L) [10].

### 2.2. Genome Sequencing, Assembly, and Annotation

The genome of the G. lingzhi strain was de novo sequenced by high-throughput Illumina HiSeqX-Ten and PacBio RSII long-read sequencing platforms (PacBio P6-C4) at Shanghai OE Biotech. Co., Ltd. in our previous studies [9]. These two sequencing platforms were widely used in the genome sequencing analysis of many species. Briefly, FALCON (version 0.7.0) was used for the de novo assembly of contigs [11]. Further, the PacBio-corrected contigs with accurate Illumina short reads and generating the genome assembly was performed with Pilon (v 1.23) [12]. Finally, BUSCO 3.1.0 was used with comparison to the lineage dataset fungi_odb9 to evaluate the genome assembly [13]. Homologous comparison and de novo prediction were used to annotate the G. lingzhi genome sequences according to the previous description [14]. 

### 2.3. Identification of CAZymes and Mating Type (MAT) Genes

The BLASTP search (covered fraction ratio ≥ 0.2; e-value ≤ 1 × 10^−6^; maximum hit number = 500) of dbCAN HMMs 6.0 and Hidden Markov Model (HMM) search (default cutoff threshold) were used for the analysis of the Carbohydrate-active enzymes (CAZymes) family [15]. The final CAZyme annotation was obtained from the common results of these two methods.

By using TBLASTN and MAT-A genes from *G. lucidum,* the MAT-A genes in *G. lingzhi* were identified including homeodomain type 1 and homeodomain type 2 mating-type genes (HD1 and HD2 genes) [8]. The same method is used to identify the mitochondrial intermediate peptidase gene (mip) in *G. lingzhi*. The pheromone receptor genes (MAT-B) in *G. lingzhi* were identified by the Swissprot annotation with the keyword “pheromone receptor”.

### 2.4. Ganoderic Acid Extraction and Mass Spectrometry Analysis

Fermentation mycelium samples of *G. lingzhi* were freeze-dried by a vacuum freeze dryer (Scientz-100F). The sample was ground into powder and then dissolved in a 70% methanol solution. Vortex extraction was performed 6 times (30 s every 30 min), and then overnight extraction was performed at 4 °C. After centrifugation (12,000 rpm, 10 min), the supernatant is used for subsequent UPLC–MS/MS analysis.

A UPLC–ESI–MS/MS system (UPLC, Shim-pack UFLC SHIMADZU CBM A system) was used for the analysis of GAs. The UPLC C18 column (1.8 µm, 2.1 mm × 100 mm) was used to sample diversion in water (0.1% formic acid)-acetonitrile (0.1% acetic acid) solution by gradient at 40 °C, 0.4 mL/min flow rate. 

Mass spectrometry was performed using an AB4500 Q TRAP UPLC/MS/MS System, equipped with an ESI Turbo Ion-Spray interface. The ESI source, turbo spray at 550 °C; 5500 V and −4500 V at positive and negative ion mode respectively. Gas I, II, and curtain gas were set at 50, 60, and 25.0 psi, respectively. Triple quadrupole scans were acquired as Multiple Reaction Monitoring (MRM) experiments (nitrogen as the medium), and MRM transitions were monitored.

### 2.5. RNA-Seq

*G. lingzhi* fresh fermented mycelium (six biological repeats) was used to perform RNA extraction by TRIzol Reagent (Life Technologies, Carlsbad, CA, USA) according to the manufacturer’s instructions. The cDNA library was constructed following the instruction manual of the NEBNext Ultra RNA Library Prep Kit (NEB, E7530) and NEBNext Multiplex Oligos (NEB, E7500).

The library was sequenced using the Illumina HiSeq platform. Transcriptome analysis was performed using *G. Lingzhi* reference genome-based read mapping. Raw data (raw reads) of fastq format were first processed through in-house perl scripts. Clean data (clean reads) were obtained by using Tophat2 software to remove the low-quality reads from the raw data, such as unknown nucleotides >5%, reads containing ploy-N or a Q20 <20% (error rates < 1%) [16]. The aligned records were further examined to remove potential duplicate molecules. 

Discovery of novel transcripts and genes was achieved by StringTie on the base of the reference genome [17]. The transcripts without annotations compared with original genome annotations are defined as novel transcripts. The transcripts containing only one exon or short transcripts (less than 150 bp) were excluded. Novel genes were annotated by DIAMOND [18] against databases including NR [19], Swiss-Prot [20], COG [21], KOG [22], and KEGG [23]. 

StringTie was applied to assemble the mapped reads generated by Hisat2 [24]. Asprofile [25] was employed to predict alternative splicing events in each sample and sort them into 12 types.

## 3. Results and Discussion

### 3.1. The Genome Characteristics of G. lingzhi

Using a whole-genome shotgun sequencing strategy, the genome of *G. lingzhi* was sequenced [9]. Genome sequences of 49.15 Mb were assembled into 30 scaffolds, with an average length of 1.65 Mb. An amount of 13,125 protein-coding genes were predicted, with an average mRNA and CDS length of 1.95 kb and 1.45 kb, respectively. On average, each predicted gene contained 6.04 exons, with an average length of 240.48 bp (Appendix A). 

Further, we identified 7,275,816 bp repeat sequences, accounting for 14.80% of the genome (Table 1). Of them, long terminal repeats (LTRs), DNA transposons, and simple repeats were the main categories, accounting for 5.71%, 2.06%, and 0.64% of the genome, respectively. Short/long interspersed nuclear elements (SINEs/LINEs) and satellite DNA comprised less than 0.10% of the genome. In addition, 178 noncoding RNAs containing 141 tRNAs, 22 snRNAs, 13 rRNAs, and 2 sRNAs with a total length of 32,535 bp were identified (Table 1).

### 3.2. Gene Annotation

Eight public databases were used to annotate the function of predicted genes, including UniProt, Gene Ontology (GO), Kyoto Encyclopedia of Genes and Genomes (KEGG), Pfam, RefSeq, NCBI non-redundant, NCBI clusters of orthologous groups of proteins (COG), and Pathway. Overall, 12,802 genes (accounting for 97.54% of all genes) were annotated to at least one function. A total of 6464 genes (49.25%), 9166 genes (69.84%), 4634 genes (35.31%), 12,762 genes (97.23%), 6406 genes (48.81%), 4734 genes (36.07%), 2898 genes (22.08%), and 1274 genes (9.71%) were annotated with UniProt, Pfam, Refseq, NCBI nr, GO, KEGG, Pathway, and COG, respectively. The annotation results were combined and shown in Appendix A.

### 3.3. GO and KEGG Analysis

The GO enrichment analysis showed that transmembrane transport was the most representative term, followed by intracellular protein transduction and cell division in biological processes (Figure 1A, left). In cell components, the nucleus was the most representative term, followed by cytosol and the integral component of the membrane (Figure 1A, centre). In molecular function, ATP binding was the most representative term, followed by metal ion binding and RNA binding (Figure 1A, right). The KEGG analysis revealed that all the predicted genes were enriched in five classes with 31 pathways in total (Figure 1B; organismal systems with 9 pathways, metabolism with 12 pathways, genetic information processing with 4 pathways, environmental information processing with 2 pathways, and cellular processes with 4 pathways). Similar to the KEGG annotation, metabolic and biosynthesis categories in COG were highly enriched (Figure 1C), such as carbohydrate transport and metabolism, lipid transport and metabolism, and secondary metabolic biosynthesis (Figure 1C Group G, I, and Q).

### 3.4. Identification of CAZymes

In nature, *G. lingzhi* often grows on rotten or living wood by degrading plant cell walls with carbohydrate-active enzymes (CAZymes). A total of 510 *G. lingzhi* genes could be assigned to CAZyme families (Figure 2, Appendix A) as defined in the CAZy database by a Hidden Markov Model (HMM) search [15]. In detail, 254 glycoside hydrolases (GHs), 79 glycosyl transferases (GTs), 9 polysaccharide lyases (PLs), 117 auxiliary activities (AAs), 47 carbohydrate esterases (CEs), and 4 carbohydrate-binding modules (CBMs) were searched. In the wide spectrum of GH families, many genes for digesting cellulose (such as GH5 with 20 genes), hemicellulose (such as GH3, GH10, and GH43 with 13, 10, and 15 genes, respectively), and chitin (such as GH18 with 21 genes) were found in the *G. lingzhi* genome (Figure 2).

The CAZyme profile in *G. lingzhi* was also compared to those of 14 other fungi containing two *G. lucidum* species (Appendix A). The global number of GTs, GHs, PLs, CEs, and CBMs of *G. lingzhi* was comparable with those of two *G. lucidum* species, Xiangnong No. 1 [6] and 260125-1 [8], and other white-rot fungi, such as *Lentinula edodes* [26] and *P. ostreatus* [27], but larger than those of *Hericium erinaceus* [14] and brown-rot fungi *Serpula lacrymans* [28]. In addition, the number of CBMs of *G. lingzhi* was significantly less than that of *G. lucidum* and the other species except for *H. erinaceus* (Appendix A), indicating different CAZyme activity properties between *G. lingzhi* and *G. lucidum* in plant cell wall degradation.

### 3.5. Mating Gene Loci in G. lingzhi

Mating is the crucial step in the fruiting and sexual reproduction of mushroom-forming fungi and is governed by mating genes that are located at distinct loci called mating type (MAT) loci [29]. *Ganoderma* spp. are tetrapolar species with two distinct mating type loci (MAT-A and MAT-B) [30]. The MAT-A locus contains genes encoding homeodomain transcription factors, including homeodomain type 1 and type 2 mating-type genes (HD1 and HD2 genes). As shown in Figure 3A, these two genes are adjacent to one another in opposite orientations on scaffold 1 and adjacent to mitochondrial intermediate peptidase (mip) and beta flanking gene (*β-fg*), as observed in other basidiomycete species [31,32]. The MAT-B locus genes encode pheromones and pheromone receptors. Eleven potential pheromone receptor genes were identified and clustered on scaffold 9, which is homologous to the STE3 gene (Figure 3B). However, no pheromone precursor genes were found in the proximity ~20 kb flanking region of these pheromone receptors, which is similar to reports in the medicinal mushroom *Hericium erinaceus* [14]. 

### 3.6. The Pathway of Ganoderic Acid Biosynthesis

At present, the synthetic pathway of GAs, which is an important medicinal component found in *Ganoderma* spp., is still not well elucidated. GAs are synthesized via the mevalonate (MVA) pathway, which is conserved in all eukaryotes. A total of 13 genes encoding 11 enzymes existed in the terpenoid backbone biosynthesis (map00900) pathway (Appendix A). The genome of *G. lingzhi* contained two farnesyl diphosphate synthases and two squalene monooxygenases, which is different from that of *G. lucidum* which contained two acetyl-CoA acetyltransferase genes and two farnesyl diphosphate synthase genes [8]. The catalytic synthesis of lanosterol from acetyl coenzyme A by the MVA pathway was well-studied. The following catalytic synthesis of GAs from lanosterol was performed by the cytochrome P450 superfamily (CYPs), which includes various complex and unclear oxidation, reduction, and acylation modification reactions. As shown in Appendix A, 145 CYP coding genes have been annotated, which requires further research to clarify their functions in GA biosynthesis.

### 3.7. Identification of GAs by Mass Spectrometry

The GAs were further identified by ultrahigh-performance liquid chromatography-electrospray ionization-tandem mass spectrometry (UPLC-ESI-MS/MS, fragmentation ions of MS spectra are shown in Appendix A). We tentatively identified and deduced the possible structure of 20 main GA constituents, including 8 GAs of C20, 5 GAs of C34, 4 GAs of C32, and 1 GAs of C24, C33, and C36 each (Figure 4 and Table 2). Seventeen ganoderic acids were identified in *G. lingzhi* including well-known ganoderic acids A, T, Me, and C2. Some less-known ganoderic acids, AP2 and GS-3, were also identified in *G. lingzhi* which were previously found in *Ganoderma applanatum* [33] and *Ganoderma sinense* [34], respectively. In addition, we identified ganodermanontriol, ganoderenic acid C, and a new special GA, ganochlearic acid A (C_24_H_34_O_5_), which is a rearranged hexanorlanostane triterpenoid featuring a γ-lactone ring (A ring) and a five-membered carbon ring (B ring) (Figure 4 compound **20**). The A ring is connected with a B ring through a single bond (C-5−C-10) unambiguously deduced in *Ganoderma cochlear* [35]. 

It has been reported that there are more than 150 kinds of GAs isolated from *Ganoderma*, which are mainly from the fruiting body and spore [36,37]. The GA in the fermentation mycelia of *Ganoderma* is still unclear. Only 19 kinds of GAs were found in the mycelia of *Ganoderma* [4]. In this work, we first identified and deduced the possible structures of 20 main GA constituents by UPLC-ESI-MS/MS from *G. lingzhi mycelia*, including a new special ganochlearic acid A.

### 3.8. RNA-Seq Analysis

Further, we analyzed transcripts and alternative splicing via RNA sequencing (RNA-Seq). For deep RNA-Seq, we used the Illumina paired-end sequencing strategy to assemble six repeat *G. lingzhi* RNA libraries. The clean reads and mapping ratio against the reference genome of each repeat ranged from 21.11 to 27.41 million and 92.74% to 92.91%, respectively (Figure 5A,B). By comparing with the original annotation of the genome, a total of 3996 new transcripts were found in the RNA-Seq. The annotation results of new transcripts are shown in Appendix A, and the novel transcript sequence information has been deposited at SRR17081370.

Then, ASprofile was employed to predict alternative splicing (AS) events in each sample. More than 30,000 AS events were predicted in each sample of six replicates and sorting them into 12 types (details shown in Appendix A). As shown in Figure 6A, alternative 5′ first exon (transcription start site, TSS) splicing and alternative 3′ last exon (transcription terminal site, TTS) splicing were the most frequent AS events. Multi-skipped exon (MSKIP), approximate multi-intron retention (XMIR), and approximate multi-exon skipped exon (XMSKIP) were the AS events with less frequency. Venn analysis showed that 9276 genes were predicted to have AS events in six replicates (Figure 6B). The 9276 genes were enriched for functional categories involved in protein processing, endocytosis, and metabolic activities by KEGG. In particular, pyrimidine metabolism (*p*-value = 3.62 × 10^−2^), arginine and proline metabolism (*p*-value = 6.58 × 10^−2^), citrate cycle (*p*-value = 7.065 × 10^−2^), mismatch repair (*p*-value = 1.25 × 10^−1^) were enriched (Figure 6C and Appendix A). In addition, 21 and 3 AS event genes were enriched in steroid biosynthesis (*p*-value = 9.90 × 10^−1^) and sesquiterpenoid and triterpenoid biosynthesis (*p*-value = 7.54 × 10^−1^) pathways, respectively (Appendix A), which provided a reference for further study on the regulation mechanism of AS involved in GAs biosynthesis. 

## 4. Conclusions

In summary, we characterized the *G. lingzhi* genome with a length of 49.15-Mb, consisting of 30 scaffolds, encoding 13,125 predicted genes, which were sequenced with the combination of the Illumina HiSeq X-Ten and PacBio RSII strategy. We identified 519 CAZymes and analyzed the difference in CAZymes genes of *G. lingzhi* with those of other fungi. In addition, the mating gene loci and ganoderic acid biosynthesis pathway were characterized. Then, 20 ganoderic acids were first identified from the mycelia of *G. lingzhi* by mass spectrometry, including a new special ganochlearic acid A. Furthermore, 3996 novel transcripts were discovered and 9276 genes were predicted to have the possibility of alternative splicing from RNA-Seq data. Collectively, this study develops foundational genomic, transcriptomic, and ganoderic acid bioactive compound resources that can be used in further breeding, pharmacological research, and molecular genetic analyses in *G. lingzhi*.

## Figures and Tables

**Figure 1 jof-08-01257-f001:**
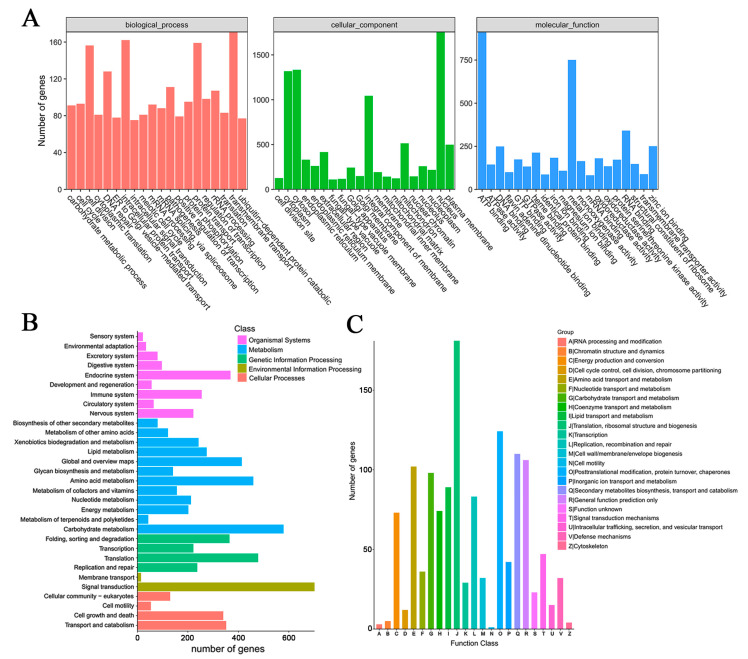
Function annotation of predicted genes in *G. lingzhi.* (**A**) The GO function annotation of predicted genes in *G. lingzhi*. Biological processes, cell components, and molecule functions are three categories of functional analysis. Counts for each category represent the total associated terms in the database with the query gene list. Terms with a *p*-value < 0.05 are statistically significant. The 20 most significantly enriched terms were listed, and the number of involved genes in a term is shown on the left y-axis. (**B**) KEGG analysis of predicted genes in *G. lingzhi*. Enriched KEGG pathways are clustered into five categories of cellular processes, environmental information processing, genetic information, processing, metabolism, and organismal systems. The number of involved genes in a term is shown on the left x-axis. (**C**) Distribution of genes in different COG function classifications of *G. lingzhi*. The number of involved genes in a term is shown on the left y-axis.

**Figure 2 jof-08-01257-f002:**
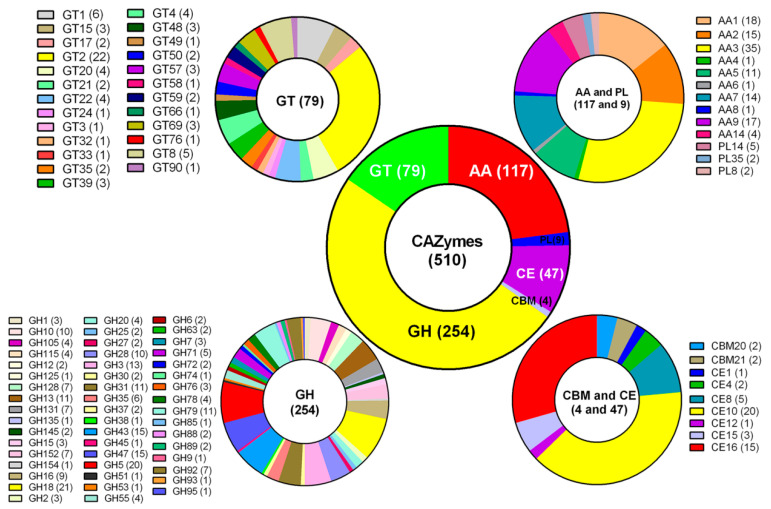
The number of carbohydrate-active enzyme genes (CAZymes) in *G. lingzhi*. GH: glycoside hydrolase; GT: glycosyl transferase; PL: polysaccharide lyase; CE: carbohydrate esterase; CBM: carbohydrate-binding module; AA: auxiliary activity. The number of involved genes in a term is shown in brackets.

**Figure 3 jof-08-01257-f003:**
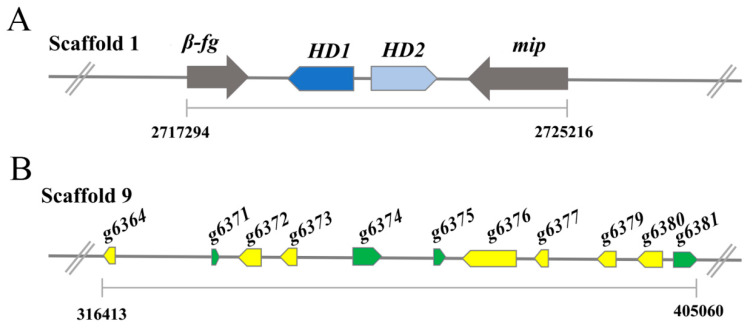
Analysis of the *G. lingzhi* MAT-A and MAT-B gene locus. (**A**) The MAT-A includes homeodomain type 1 and type 2 mating-type genes (HD1 and HD2 genes). *mip*: mitochondrial intermediate peptidase, *β-fg*: beta flanking gene. (**B**) The MAT-B locus genes encode eleven potential pheromone receptor genes.

**Figure 4 jof-08-01257-f004:**
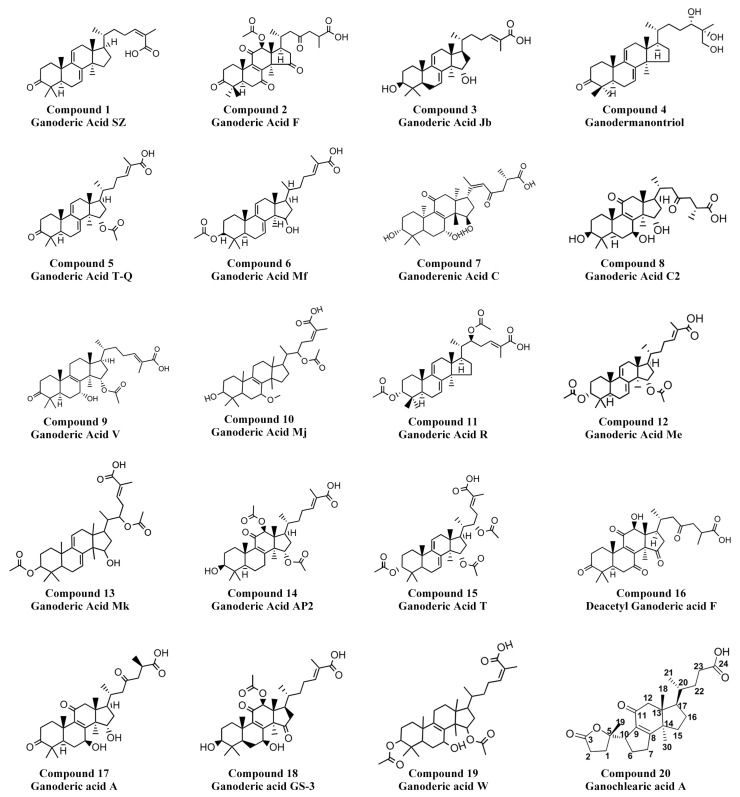
Chemical structures of the identified ganoderic acids.

**Figure 5 jof-08-01257-f005:**
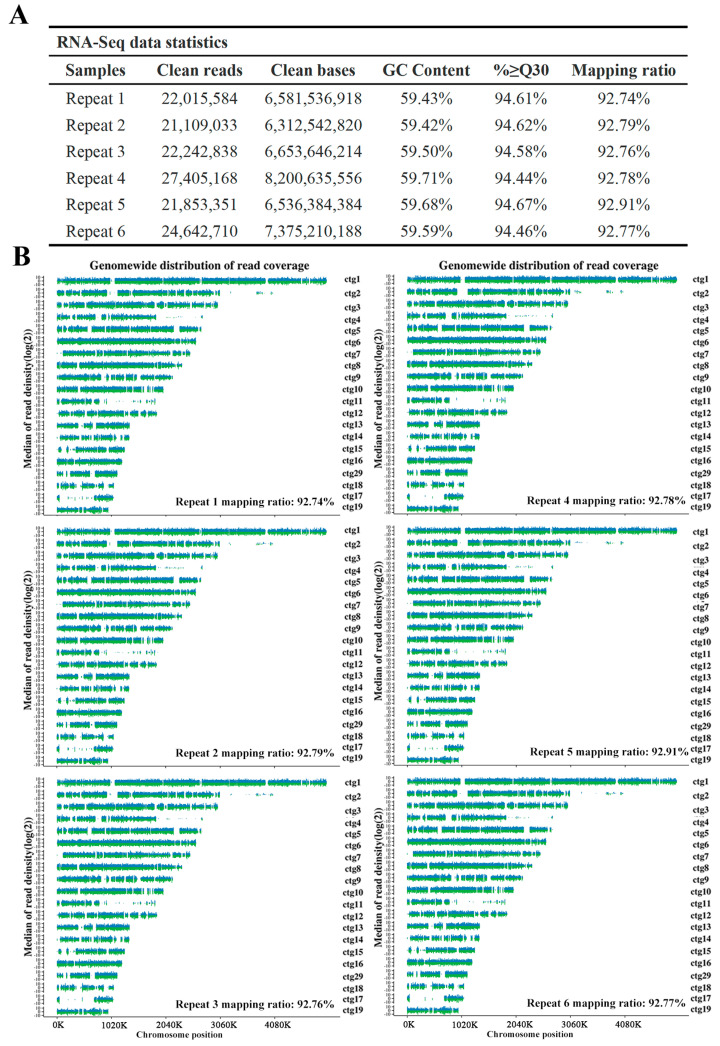
Summary of transcriptome sequencing data. (**A**) The statistics of RNA sequencing data. ≥Q30%: Percentage of bases with a Q-score no less than Q30. (**B**) Distribution of mapped reads on reference genome: position and depth. *X*-axis: Position on the chromosome; *Y*-axis: Log2 of coverage depth (coverage depth was defined as reads counted within a chromosome window of 10 kb in length); Blue represents + strand and green represents − strand.

**Figure 6 jof-08-01257-f006:**
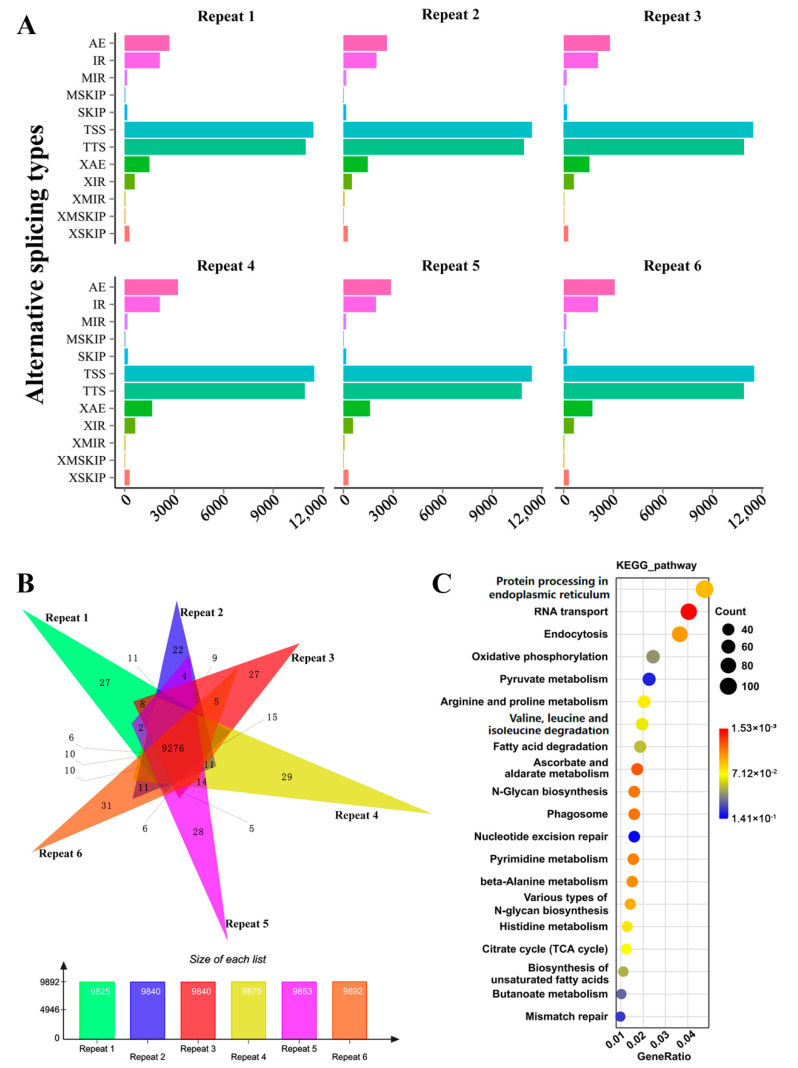
Alternative splicing analysis. (**A**) Statistics of alternative splicing events. *X*-axis: Number of transcripts in specific alternative splicing type; *Y*-axis: 12 alternative splicing types. TSS: alternative 5′ first exon (transcription start site, TSS) splicing; TTS: alternative 3′ last exon (transcription terminal site, TTS) splicing; SKIP: single exon skipping; XSKIP: approximate SKIP; MSKIP: multi-SKIP; XMSKIP: approximate MSKIP; IR: single intron retention; XIR: approximate IR; MIR: multi-IR; XMIR: approximate MIR; AE: alternative exon ends; XAE: approximate AE. (**B**) Venn analysis the alternative splicing genes in six repeat RNA-Seq. (**C**) KEGG analysis of the alternative splicing genes.

**Table 1 jof-08-01257-t001:** Characteristics of the repeat sequences and non-coding RNA of *G. lingzhi*.

Repeat Sequences			
Item	Number	Length (bp)	Coverage
SINE	6	488	0.00%
LINE	121	22,417	0.05%
LTR	1849	2,806,397	5.71%
DNA transposon	538	1,013,038	2.06%
Satellite DNA	26	2435	0.00%
Simple repeat	7431	313,771	0.64%
Low complexity	1008	51,127	0.10%
Other	42	10,370	0.02%
Unknown	6091	3,055,773	6.22%
Total	17,112	7,275,816	14.80%
**Non-Coding RNA**			
**Class**	**Number**	**Total Length (bp)**	**Mean Length (bp)**
rRNA	13	17,274	1328
sRNA	2	709	354
snRNA	22	2869	130
tRNA	141	11,683	82
Total	178	32,535	182

**Table 2 jof-08-01257-t002:** Identification of possible ganoderic acids from *G. lingzhi*.

NO.	Compounds	Molecular Wght (Da)	Q1, *m*/*z* (Da)	Fragmentation, *m*/*z* (Da)
1	Ganoderic Acid SZ (C_30_H_44_O_3_)	452.3	453.3 ([M+H]^+^)	435.3, 185.1, 187.1, 239.2, 173.1, 201.2, 225.2
2	Ganoderic Acid F (C_30_H_46_O_3_)	454.3	455.3 ([M+H]^+^)	437.3, 229.2, 299.3, 123.1, 201.2, 135.1
3	Ganoderic Acid Jb (C_30_H_46_O_4_)	470.3	471.3 ([M+H]^+^)	435.3, 453.3, 201.2, 187.1, 471.3, 159.1, 175.1
4	Ganodermanontriol (C_30_H_48_O_4_)	472.4	473.4 ([M+H]^+^)	329.2, 455.4, 415.3, 243.2, 261.2, 437.3, 189.1
5	Ganoderic Acid T-Q (C_32_H_46_O_5_)	510.3	511.3 ([M+H]^+^)	433.3, 493.3, 311.2, 451.3, 293.2, 399.3, 337.2
6	Ganoderic Acid Mf (C_32_H_48_O_5_)	512.4	513.4 ([M+H]^+^)	435.3, 495.3, 201.2, 295.2, 203.2, 453.3, 133.1
7	Ganoderenic Acid C (C_30_H_44_O_7_)	516.3	517.3 ([M+H]^+^)	371.3, 499.3, 399.3, 463.3, 481.3, 353.2, 381.2
8	Ganoderic Acid C2 (C_30_H_46_O_7_)	518.3	519.3 ([M+H]^+^)	355.3, 483.3, 501.3, 465.3, 447.3, 373.3
9	Ganoderic Acid V (C_32_H_48_O_6_)	528.3	529.3 ([M+H]^+^)	469.3, 243.2, 423.3, 329.2, 451.3, 369.3, 355.3
10	Ganoderic Acid Mj (C_33_H_52_O_6_)	544.4	545.2 ([M+H]^+^)	527.3, 449.3, 467.3, 431.3, 353.2, 421.3, 327.2
11	Ganoderic Acid R (C_34_H_50_O_6_)	554.4	555.4 ([M+H]^+^)	435.3, 495.3, 201.2, 187.1, 145.1, 239.2, 341.2
12	Ganoderic Acid Me (C_34_H_50_O_6_)	554.4	555.4 ([M+H]^+^)	435.3, 495.3, 295.2, 201.2, 203.2, 187.1, 189.2
13	Ganoderic Acid Mk (C_34_H_50_O_7_)	570.4	571.4 ([M+H]^+^)	433.3, 451.3, 493.3, 201.2, 511.3, 293.2, 339.2
14	Ganoderic Acid AP2 (C_34_H_50_O_8_)	586.4	587.4 ([M+H]^+^)	587.4, 569.4, 491.3, 509.3, 431.3, 395.3, 409.3, 463.3
15	Ganoderic Acid T (C_36_H_52_O_8_)	612.4	613.4 ([M+H]^+^)	433.3, 493.3, 553.4, 201.2, 293.2, 451.3, 227.2
16	Deacetyl Ganoderic acid F (C_30_H_40_O_8_)	528.3	527.3 ([M-H]^−^)	509.6, 479.4, 465.5, 435.6, 365.8, 315.2, 299.4
17	Ganoderic acid A (C_30_H_44_O_7_)	516.3	515.3 ([M-H]^−^)	497.5, 453.6, 435.5, 355.3, 337.3, 299.5, 285.5
18	Ganoderic acid GS-3 (C_32_H_46_O_8_)	558.3	557.3 ([M-H]^−^)	539.6, 497.4, 453.5, 435.6, 303.4, 287.4
19	Ganoderic acid W (C_34_H_52_O_7_)	572.4	571.4 ([M-H]^−^)	553.6, 523.8, 509.3, 479.3, 465.4, 419.6, 345.5, 303.5, 285.4
20	Ganochlearic acid A (C_24_H_34_O_5_)	402.5	401.2 ([M-H]^−^)	357.4, 329.4, 313.3, 287.4

## Data Availability

The assembly genome in this paper is associated with NCBI BioProject: PRJNA738334. This Whole Genome Shotgun project has been deposited at DDBJ/ENA/GenBank under the accession JAPJYM000000000. The version described in this paper is version JAPJYM010000000. Raw sequencing data for PacBio RSII reads and Illumina HiSeq X-Ten reads of the genome have been deposited in the NCBI Sequence Read Archive under accession no. SRR22226938 and SRR22226939, respectively. Six repeat RNA raw sequencing data have been deposited in NCBI’s Sequence Read Archive under accession no. SRR22226940-45. The genome sequencing information and transcripts sequencing information can be downloaded from SRR14933280 and SRR17081370.

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
