# Peer review of "Characteristics of the Genome, Transcriptome and Ganoderic Acid of the Medicinal Fungus Ganoderma lingzhi"

_jof, 2022, doi:10.3390/jof8121257_

Round 1

Reviewer 1 Report

Authors present the research on genome and transcriptome in Ganoderma lingzhi. 

Reviewer could not open the zipped file downloaded from ‘Original Images for Blots/Gels’. Therefore reviewer commented the manuscript based on the information in the main text.

First of all, reviewer cannot find any Genbank accession ID for assembled genome sequence. 

Furthermore, original information for PacBio reads. All SRA run IDs in ‘Data availability’ (SRR14933280, SRR14933281, SRR17081370) were Illumina read IDs. Original PacBio reads must be submitted to SRA also.

These information must be submitted to the public databases before the review process.

Author Response

1.Authors present the research on genome and transcriptome in Ganoderma lingzhi. Reviewer could not open the zipped file downloaded from ‘Original Images for Blots/Gels’. Therefore reviewer commented the manuscript based on the information in the main text. Response: This research paper does not involve ‘Original Images for Blots/Gels’. 2.First of all, reviewer cannot find any Genbank accession ID for assembled genome sequence. Response: Thanks for the reviewer's suggestion. The assembled genome can be downloaded from GenBank under the project accession PRJNA71455. Detailed informations were added to the Data availability Section (Please see lines 332-338) 3.Furthermore, original information for PacBio reads. All SRA run IDs in ‘Data availability’ (SRR14933280, SRR14933281, SRR17081370) were Illumina read IDs. Original PacBio reads must be submitted to SRA also.These information must be submitted to the public databases before the review process. Response: The original PacBio RSII reads and Illumina HiSeq X-Ten reads of genome have been submitted to SRA, SRR22226938 and SRR22226939, respectively. Detailed informations were added to the Data availability Section (Please see lines 332-338)

Reviewer 2 Report

jof-1967920

The manuscript shows an interesting topic of the characteristics of the genome, transcriptome and various ganoderic acid constituents formation of the medicinal fungus Ganoderma lingzhi. It is quite an interesting approach. The authors carried out the sequencing strategies, gene function analysis, and annotated genes involved in lignocellulose degradation and sexual recognition as well as ganoderic acid biosynthesis. The identification of possible structures of 20 main ganoderic acd constituents was confirmed by mass spectrometry. The abstract is good and complete; the materials and methods are comprehensive, and the results produced are novel. The paper is very good in terms of scientific content, and it is important since it is based on empirical work.

If authors can make revisions and provide a reasonable explanation based on scientific facts and issues, I think that the work should be published with regard to the novelty, systematic presentation, and discussion of the results. I have a few comments, which should be addressed prior to publication.

1. The research on ganoderic acid and its constituents of Ganoderma lingzhi or other mushrooms needs to be introduced in the introduction part, especially the role of ganoderic acid as a key bioactive metabolite.

2. Lines 59-64: In addition to Illumina HiSeq X-Ten, have the authors tried to identify the genome sequencing of G. lingzhi via “SOLiD” searches optimized for short sequences (typically 50 bp)?

3.  What was the rationale for selecting the choice of sequencing platform in this study?

4. The alignments of the characteristics of the genome, transcriptome and various ganoderic acid constituents formation of G. lingzhi should be emphasized and discussed and should be stated in the conclusions.

5. The suggestions concerning the improvement of the manuscript and some corrections of typographical and spelling errors are provided in the attached file.

Line 12: The font size is inappropriate.

Line 305: scafffolds should be scaffolds.

Line 312: Please provide author contribution statements.

Line 325: Reference should be References

Line 326 and onward: Please check font style throughout the references.

Author Response

The manuscript shows an interesting topic of the characteristics of the genome, transcriptome and various ganoderic acid constituents formation of the medicinal fungus Ganoderma lingzhi. It is quite an interesting approach. The authors carried out the sequencing strategies, gene function analysis, and annotated genes involved in lignocellulose degradation and sexual recognition as well as ganoderic acid biosynthesis. The identification of possible structures of 20 main ganoderic acd constituents was confirmed by mass spectrometry. The abstract is good and complete; the materials and methods are comprehensive, and the results produced are novel. The paper is very good in terms of scientific content, and it is important since it is based on empirical work. Response: Special thanks to the reviewer for the positive evaluation. If authors can make revisions and provide a reasonable explanation based on scientific facts and issues, I think that the work should be published with regard to the novelty, systematic presentation, and discussion of the results. I have a few comments, which should be addressed prior to publication. 1.The research on ganoderic acid and its constituents of Ganoderma lingzhi or other mushrooms needs to be introduced in the introduction part, especially the role of ganoderic acid as a key bioactive metabolite. Response: Thanks. According to the reviewer's suggestion, the role of ganoderic acid as a key bioactive metabolite of Ganoderma have been introduced in the introduction part (Please see lines 41-45). 2. Lines 59-64: In addition to Illumina HiSeq X-Ten, have the authors tried to identify the genome sequencing of G. lingzhi via “SOLiD” searches optimized for short sequences (typically 50 bp)? Response: In this study, second-generation sequencing technology, Illumina HiSeq X-Ten, and third-generation sequencing technology, PacBio RSII, were used to identify the genome sequencing of G. lingzhi. And Pilon (v 1.23) was utilized to correct the PacBio-corrected contigs with accurate Illumina short reads and to generate the genome assembly of G. lingzhi. We regret that we did not use another second-generation sequencing technology, SOLiD, to optimize short sequences. 3.What was the rationale for selecting the choice of sequencing platform in this study? Response: The Illumina and PacBio sequencing platform were widely used in genome sequencing analysis of many species[1-5]. Herein, with the combination of Illumina HiSeq X-Ten and PacBio RSII sequencing strategy, the genome of G. lingzhi was de novo sequenced and assembled. And the rationale for selecting the choice of sequencing platform in this study were added to lines 74-75. 1.Jenjaroenpun P, Wongsurawat T, Pereira R, Patumcharoenpol P, Ussery DW, Nielsen J, Nookaew I. Complete genomic and transcriptional landscape analysis using third-generation sequencing: a case study of Saccharomyces cerevisiae CEN.PK113-7D. Nucleic Acids Res. 2018 Apr 20;46(7):e38. doi: 10.1093/nar/gky014. PMID: 29346625; PMCID: PMC5909453. 2.Xie SY, Ma T, Zhao N, Zhang X, Fang B, Huang L. Whole-genome sequencing and comparative genome analysis of Fusarium solani-melongenae causing fusarium root and stem rot in sweetpotatoes. Microbiol Spectr. 2022 Aug 31;10(4):e0068322. doi: 10.1128/spectrum.00683-22. Epub 2022 Jul 7. PMID: 35863027; PMCID: PMC9430127. 3.Gong W, Wang Y, Xie C, Zhou Y, Zhu Z, Peng Y. Whole genome sequence of an edible and medicinal mushroom, Hericium erinaceus (Basidiomycota, Fungi). Genomics. 2020 May;112(3):2393-2399. doi: 10.1016/j.ygeno.2020.01.011. Epub 2020 Jan 21. PMID: 31978421. 4.Chen S, Xu J, Liu C, Zhu Y, Nelson DR, Zhou S, Li C, Wang L, Guo X, Sun Y, Luo H, Li Y, Song J, Henrissat B, Levasseur A, Qian J, Li J, Luo X, Shi L, He L, Xiang L, Xu X, Niu Y, Li Q, Han MV, Yan H, Zhang J, Chen H, Lv A, Wang Z, Liu M, Schwartz DC, Sun C. Genome sequence of the model medicinal mushroom Ganoderma lucidum. Nat Commun. 2012 Jun 26;3:913. doi: 10.1038/ncomms1923. PMID: 22735441; PMCID: PMC3621433. 5.Kües U, Nelson DR, Liu C, Yu GJ, Zhang J, Li J, Wang XC, Sun H. Genome analysis of medicinal Ganoderma spp. with plant-pathogenic and saprotrophic life-styles. Phytochemistry. 2015 Jun;114:18-37. doi: 10.1016/j.phytochem.2014.11.019. Epub 2015 Feb 11. PMID: 25682509. 4. The alignments of the characteristics of the genome, transcriptome and various ganoderic acid constituents formation of G. lingzhi should be emphasized and discussed and should be stated in the conclusions. Response: Thanks. According to the reviewer's suggestion, we emphasized and discussed the alignments of the characteristics of the genome, transcriptome and various ganoderic acid constituents formation of G. lingzhi in lines 198-206, 218-221, 248-256, 259-264, and the conclusion section was rewritten in lines 308-319. 5. The suggestions concerning the improvement of the manuscript and some corrections of typographical and spelling errors are provided in the attached file. Response: Thanks. We have revised them one by one according to the suggestions. 5.1 Line 12: The font size is inappropriate. Response: We revised the font size of “Correspondence:” in line 12. 5.2 Line 305: scafffolds should be scaffolds. Response: According to the reviewer's suggestion, we revised “scafffolds” to “scaffolds” in line 309 5.3 Line 312: Please provide author contribution statements. Response: According to the reviewer's suggestion, we provided the Author Contributions statements (Please see lines 320-324). 5.4 Line 325: Reference should be References Response: According to the reviewer's suggestion, we revised “Reference” to “References” in line 341. 5.5 Line 326 and onward: Please check font style throughout the references. Response: According to the reviewer's suggestion, we carefully checked the font style throughout the references.

Reviewer 3 Report

The manuscript describes a substantial body of work investigating the genome, transcriptome and ganoderic acid biosynthesis of an important medicinal fungus, Ganoderma lingzhi, and will be of interest to workers in this field of research, however the results appear to be incomplete. There is no accession number provided for the genomic data. Is it  publicly available?  Tables 1 and 2 (lines 138 and 140) are missing. The supplementary material also needs to be available for review. Some of the figures are illegible, e.g. Figure 1 has a very small font that needs to be increased in size, Figure 5 also has small font and is blurry. It should not be a major task to rectify these issues. 

Minor points - lingzhi is misspelt on line 111, scaffolds is misspelt on line 305.

Author Response

The manuscript describes a substantial body of work investigating the genome, transcriptome and ganoderic acid biosynthesis of an important medicinal fungus, Ganoderma lingzhi, and will be of interest to workers in this field of research, however the results appear to be incomplete. Response: Thanks to the reviewer for the evaluation. 1.There is no accession number provided for the genomic data. Is it publicly available? Response: The assembled genome can be downloaded from GenBank under the project accession PRJNA71455. Detailed informations were added to the Data availability Section (Please see lines 332-338) 2.Tables 1 and 2 (lines 138 and 140) are missing. Response: We uploaded all the table data in the revised version. 3.The supplementary material also needs to be available for review. Response: We uploaded the supplementary material in the revised version. 4.Some of the figures are illegible, e.g. Figure 1 has a very small font that needs to be increased in size, Figure 5 also has small font and is blurry. It should not be a major task to rectify these issues. Response: Thanks. According to the reviewer's suggestion, we revised and increased the font size in the new Figure 1 and Figure 5. 5.Minor points - lingzhi is misspelt on line 111, scaffolds is misspelt on line 305. Response: Thanks. We revised “G. lignzhi” to “G. lingzhi” in line 111, and we revised “scafffolds” to “scaffolds” line 309.

Round 2

Reviewer 1 Report

First of all, 'PRJNA71455' is a BioProject ID, not Genbank ID. The assembly cannot be accessed from the search with term 'PRJNA71455'. The assembly linked from Bioproject entry 'PRJNA71455' is below. 

https://www.ncbi.nlm.nih.gov/assembly/GCA_000271565.1

The new assembly obtained from this work must be submitted to Genbank, and describe in the manuscript.

Author Response

Response: We are sorry to provide an incorrect a BioProject ID 'PRJNA71455'. The assembly genome in this paper is associated with NCBI BioProject: PRJNA738334. According to the reviewer's suggestion, this Whole Genome Shotgun project has been deposited at DDBJ/ENA/GenBank under the accession JAPJYM000000000. The version described in this paper is version JAPJYM010000000. These informations were added to Data Availability Section. The submission has passed the initial validation checks and has been moved to the processing queue. And before being released, the genome will be manually reviewed by indexing staff to be sure that there are no remaining errors or problems. The genomes remain at the ‘processing’ stage until they have been made public